# Development of a standardized MRI protocol for pancreas assessment in humans

John Virostko[1,2,3]*, Richard C. Craddock[1], Jonathan M. Williams[4], Taylor M. Triolo[5], Melissa A. Hilmes[6,7], Hakmook Kang[8], Liping Du[8], Jordan J. Wright[4], Mara Kinney[5], Jeffrey H. Maki[9], Milica Medved[10], Michaela Waibel[11], Thomas W. H. Kay[11,12], Helen E. Thomas[11,12], Siri Atma W. Greeley[13], Andrea K. Steck[5], Daniel J. Moore[7,14], Alvin C. Powers[4,15,16]

1 Department of Diagnostic Medicine, University of Texas at Austin, Austin, Texas, United States of America, 2 Livestrong Cancer Institutes, University of Texas at Austin, Austin, Texas, United States of America, 3 Department of Oncology, University of Texas at Austin, Austin, Texas, United States of America, 4 Division of Diabetes, Endocrinology, and Metabolism, Department of Medicine, Vanderbilt University Medical Center, Nashville, Tennessee, United States of America, 5 Barbara Davis Center for Diabetes, University of Colorado School of Medicine, Aurora, Colorado, United States of America, 6 Department of Radiology and Radiological Sciences, Vanderbilt University Medical Center, Nashville, Tennessee, United States of America, 7 Department of Pediatrics, Vanderbilt University Medical Center, Nashville, Tennessee, United States of America, 8 Department of Biostatistics, Vanderbilt University Medical Center, Nashville, Tennessee, United States of America, 9 Department of Radiology, University of Colorado School of Medicine, Aurora, Colorado, United States of America, 10 Department of Radiology, University of Chicago, Chicago, IL, United States of America, 11 Immunology and Diabetes Unit, St Vincent's Institute, Fitzroy, Victoria, Australia, 12 Department of Medicine, St. Vincent's Hospital, The University of Melbourne, Fitzroy, Victoria, Australia, 13 Section of Adult and Pediatric Endocrinology, Diabetes, and Metabolism, Kovler Diabetes Center, University of Chicago, Chicago, IL, United States of America, 14 Department of Pathology, Immunology, and Microbiology, Vanderbilt University, Nashville, Tennessee, United States of America, 15 Department of Molecular Physiology and Biophysics, Vanderbilt University, Nashville, Tennessee, United States of America, 16 VA Tennessee Valley Healthcare System, Nashville, Tennessee, United States of America

* jack.virostko@austin.utexas.edu

**Data Availability Statement:** MRI files contain protected health information (PHI) and can only be shared with IRB permission. Data are available from the Texas Data Repository (contact via

## Abstract

Magnetic resonance imaging (MRI) has detected changes in pancreas volume and other characteristics in type 1 and type 2 diabetes. However, differences in MRI technology and approaches across locations currently limit the incorporation of pancreas imaging into multi-site trials. The purpose of this study was to develop a standardized MRI protocol for pancreas imaging and to define the reproducibility of these measurements. Calibrated phantoms with known MRI properties were imaged at five sites with differing MRI hardware and software to develop a harmonized MRI imaging protocol. Subsequently, five healthy volunteers underwent MRI at four sites using the harmonized protocol to assess pancreas size, shape, apparent diffusion coefficient (ADC), longitudinal relaxation time (T1), magnetization transfer ratio (MTR), and pancreas and hepatic fat fraction. Following harmonization, pancreas size, surface area to volume ratio, diffusion, and longitudinal relaxation time were reproducible, with coefficients of variation less than 10%. In contrast, non-standardized image processing led to greater variation in MRI measurements. By using a standardized MRI image acquisition and processing protocol, quantitative MRI of the pancreas performed at multiple locations can be incorporated into clinical trials comparing pancreas imaging measures and metabolic state in individuals with type 1 or type 2 diabetes.

support@tdl.org) for researchers who meet the criteria for access to confidential data.

**Funding:** This work was supported by JDRF International (3-SRA-2019-759-M-B and 3-SRA-2015-102-M-B), the National Institute of Diabetes and Digestive and Kidney Diseases Vanderbilt Diabetes (DK20593 and DK20595), the Cain Foundation, UL1 TR002389 (Chicago Institute for Translational Medicine), and Gifts from the Kovler Family Foundation. SVI receives support from the Operational Infrastructure Support Scheme of the Victorian State Government. This work utilized REDCap which is supported by UL1 TR000445 from NCATS/NIHT. T.M.T is supported by NIDDK K12DK094712. The sponsors did not play any role in the study design, data collection and analysis, decision to publish, or preparation of the manuscript.

**Competing interests:** The authors have declared that no competing interests exist.

## Introduction

MRI plays an important role in the diagnosis and monitoring of a number of pancreatic disorders including chronic [1] and acute [2] pancreatitis, nonalcoholic fatty pancreas disease [3], and pancreatic cancer [4]. In addition, a number of quantitative MRI parameters are currently under investigation in the study of individuals with diabetes. For instance, MRI has demonstrated that pancreas volume is decreased in individuals with both type 1 [5–8] and type 2 diabetes [9,10], and may identify those at risk for developing diabetes. Longitudinal imaging in individuals with diabetes demonstrates dynamic changes in pancreas size, including declines over the natural history of type 1 diabetes [5] and increases in individuals with type 2 diabetes who respond to treatment [11]. These results suggest that MRI may have important applications in diabetes research and ultimately clinical management of the disease.

Most clinical MRI data of the pancreas is analyzed by visual inspection to identify features, such as mass lesions or edema, that are indicative of disease presence and severity. Radiomics offers a complementary approach, in which numerical features, some of which are directly related to physical or physiological properties of the imaged tissue, are extracted from the MRI data and evaluated mathematically. The advantage of this quantitative approach is that it can be sensitive to nuanced differences in images that are hard or impossible to detect visually. The clinical utility of quantitative imaging measures has been demonstrated across a number of diseases and organs of interest [12]. In the pancreas, quantitative evaluations of diffusion-weighted MRI has shown utility for characterizing pancreatic cancer [13], diagnosing pancreatitis [14], and identifying focal lesions in type 1 diabetes [5]. Two other quantitative MRI parameters, magnetization transfer ratio (MTR) and the longitudinal relaxation time (T1), have been implicated as markers of fibrosis [15,16] and may be altered in individuals with impaired glucose tolerance [17]. Measures of fat deposition in the liver have been shown to be sensitive to nonalcoholic fatty liver disease, which is a common comorbidity with type 2 diabetes [18]. Similarly, measures of pancreas fat may be associated with diabetes [19] and may correlate with therapeutic response [20].

While a variety of approaches have shown promise for pancreas imaging at single research sites, they have not been validated in multisite studies. Imaging of the brain [21] and solid tumors [22] has benefitted from multisite standardization. However, the pancreas is challenging to image, owing to its location deep within the abdomen, irregular borders, and the presence of motion and magnetic susceptibility artifacts. Thus, similar validation studies have not been performed for pancreas MRI. This represents a critical barrier for applying MRI of the pancreas to broader populations or to clinical trials. Furthermore, quantitative imaging measures can be influenced by technical factors that may vary across different types of MRI scanners, image acquisition parameters, and choice of image processing techniques. In order for quantitative parameters from pancreatic MRI to be implemented in multisite clinical trials, such as those performed by the Type 1 Diabetes TrialNet network and those in type 2 diabetes, standard image acquisition and processing protocols must be developed and validated. This study develops, validates, and makes available for universal use a multiparametric MRI protocol for pancreas imaging, which has been harmonized and assessed for reproducibility using phantoms and individuals at multiple imaging sites.

## Materials and methods

### MRI scanning protocol

The Multicenter Assessment of the Pancreas in Type 1 Diabetes (MAP-T1D) study is an international consortium of diabetes and medical imaging centers using MRI to investigate the

pancreas in individuals with type 1 diabetes. The MAP-T1D team consists of five academic centers with different MRI hardware and software; including the University of Texas at Austin, subsequently referred to as Austin; Vanderbilt University Medical Center, subsequently referred to as Nashville; Barbara Davis Center for Diabetes and University of Colorado School of Medicine, subsequently referred to as Denver; University of Chicago, subsequently referred to as Chicago; St Vincent's Institute and Hospital and the University of Melbourne, subsequently referred to as Melbourne. Two centers (Austin and Denver) were equipped with 3T Skyra scanners (Siemens, Erlangen, Germany). Scans performed in Melbourne were acquired using a 3T Prisma scanner (Siemens, Erlangen, Germany). Two centers (Nashville and Chicago) acquired images on 3T Ingenia scanners (Philips, Best, Netherlands). All Siemens scanners employed VE11C software. Of the Philips scanners, Vanderbilt employed R5.5.0.1 while Chicago employed R.5.6.1. All sites employed torso coil arrays.

The MRI protocol consisted of a three-plane localizer followed by a series of axial scans spanning the pancreas. Anatomical scans included fat-suppressed 3D T1-weighted gradient echo and T2-weighted fast spin-echo images with and without fat saturation. Diffusion-weighted MRI (DWI) was acquired in a single direction with spin-echo EPI readout and b-values of 0, 50, 200, and 800. For T1 mapping, a B1 field map was acquired to correct for transmit inhomogeneity followed by five spoiled gradient echo images with equally spaced flip angles spanning 4˚ to 20˚. A gradient echo image with and without a magnetization transfer sensitive prepulse was acquired for calculation of MTR. A 3D quantitative 6-point Dixon acquisition was collected at sites which had the requisite software on their scanner (Austin, Nashville, Chicago, Melbourne). Imaging parameters are summarized in Table 1. With the exception of the B1 map, slice thickness was 4 mm. Total time for this protocol was approximately 16 minutes of acquisition time, translating to approximately 35 minutes of total scan time.

**Table 1. MRI protocol acquisition parameters.**

| MRI Scan | 3D T1-w image | T2-w image (fat sat) | T2-w image (no fat sat) | Diffusion-weighted | B1 Map | Multi Flip Angle Spoiled GRE (T1 Mapping) | Magnetization Transfer | 3D Quantitative DIXON |
|---|---|---|---|---|---|---|---|---|
| **Orientation** | Axial | Axial | Axial | Axial | Axial | Axial | Axial | Axial |
| **Acquisition Matrix** | 256 x 208 | 256 x 208 | 256 x 208 | 128 x 104 | 128 x 104 | 128 x 104 | 128 x 104 | 160 x 132 |
| **Field of View [mm]** | 384 x 312 | 384 x 312 | 384 x 312 | 384 x 312 | 384 x 312 | 384 x 312 | 384 x 312 | 450 x 372 |
| **In Plane Resolution [mm]** | 1.5 x 1.5 | 1.5 x 1.5 | 1.5 x 1.5 | 3 x 3 | 3 x 3 | 3 x 3 | 3 x 3 | 1.4 x 1.4 |
| **Number of Slices** | 48 | 48 | 48 | 40 | 24 | 48 | 48 | 64 |
| **Slice Thickness/Slice Gap [mm]** | 4/0 | 4/0 | 4/0 | 4/0.8 | 8/0 | 4/0 | 4/0 | 4/0 |
| **TR [ms]** | 4.04 | 750 | 386 | 7600 | 14310 | 4.6 | 2000 | 9 |
| **TE [ms]** | 1.29; 2.52 | 105 | 105 | 48 | 2.06 | 1.96 | 3.58 | 1.05; 2.46; 3.69; 4.92; 6.15; 7.38 |
| **Flip Angle [degrees]** | 10 | 100 | 100 | 90 | 8 | 20; 16; 12; 8; 4 | 25 | 4 |
| **Fat Suppression** | 2 point Dixon | SPAIR | None | SPAIR | None | None | None | 6 point Dixon |
| **Miscellaneous** | | | | b-values: 0, 50, 200, 800 | | 5 flip angles | 'MTC' off & 'MTC' on | |
| **Motion Compensation** | 1 breath-hold | 2 breath-holds | 1 breath-hold | Respiratory Gated | 2 breath-holds | 1 breath-hold | None | 1 breath-hold |
| **Acquisition Time [m:s]** | 0:12 | 0:39 | 0:19 | 4:08 | 0:29 | 0:10 each; 5 acquisitions | 2:12 each; 2 acquisitions | 0:13 |

## Phantom studies

Synthetic imaging phantoms for standardizing MRI parameters were constructed at a single center (Austin) and shipped to each site for imaging (Fig 1A). These MRI phantoms were used to standardize measurements of volume, apparent diffusion coefficient (ADC), T1, MTR, and fat fraction. For volume standardization, a pancreas phantom was generated from an abdominal MRI of a 39-year-old male with no known pancreas pathology. A 3D volume of the pancreas was extracted from the abdominal MRI by freehand tracing the organ's borders using Osirix software (Pixmeo, Bernex, Switzerland). This volume was 3D printed using PLA thermoplastic and embedded in agar in a 1L Nalgene jar. For diffusion imaging, the phantom consisted of a 50 mL tube filled with deionized water chilled to 0˚C and immobilized in a 1L Nalgene jar filled with ice water, as previously described [23]. Standardization of T1 measurements employed four phantoms consisting of Gadavist-doped gelatin in 50 mL tubes with prescribed T1 values of 500 ms, 1000 ms, 1250 ms, and 1500 ms. For MTR validation, crosslinked 15% bovine serum albumin was prepared as described previously [24]. The fat fraction phantom consisted of a 50 ml tube filled with canola oil. Phantoms assessing T1, MTR, and fat fraction were all placed in the same 1L Nalgene jar filled with room temperature water prior to imaging. The spatial orientation of each phantom during scanning was identical at each site. All phantom scans were completed within 60 days of one another.

## Volunteer studies

Five healthy volunteers with no known pancreas pathology or diabetes traveled to four MAP-T1D locations in the US (Austin, Chicago, Denver, Nashville) to undergo MRI using the standardized, harmonized imaging protocol (Fig 1B). The volunteers were not imaged at Melbourne because of travel distance. Characteristics of the volunteers were: four males and one

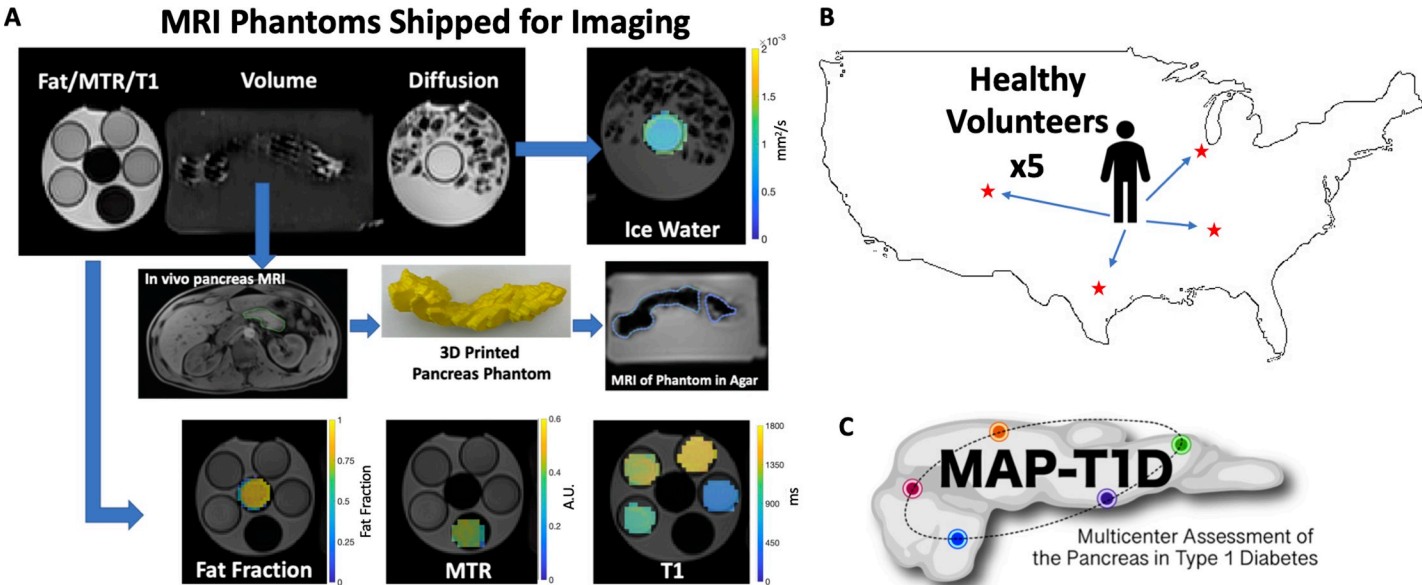

**Fig 1. Schematic of MRI data acquisition.** A) Example MRI of synthetic phantoms with calibrated properties that were shipped to five different sites for imaging. The phantom (upper left) consisted of three components. The leftmost bottle contained phantoms with canola oil, bovine serum albumin, and gadolinium-doped gelatin to validate fat fraction, MTR, and T1 measurements, respectively. Example fat fraction, MTR, and T1 maps are shown on the bottom row. The middle bottle contained a 3D printed pancreas created from an MRI of a normal volunteer pancreas and subsequently embedded in agar for imaging (middle row). The rightmost bottle contained deionized water chilled to 0˚C to validate diffusion-weighted MRI measurements. B) Five healthy volunteers traveled to four sites in the US (Austin, Chicago, Denver, and Nashville) for an MRI of the pancreas using the harmonized acquisition protocol. C) MAP-T1D study logo.

female with an average age of 37 years old (min 31, max 44) with an average BMI of $26.4 \pm 2.9$ kg/m$^2$. The time between the first and final MRI for each volunteer was 28, 28, 44, 64, and 79 days. These studies were approved by each site's Institutional Review Board and performed in accordance with the guidelines and regulations set forth by the Human Research Protections Program. Written consent was provided by each research participant.

## Image processing

Images of the phantoms and human volunteers were analyzed using the same methodology. The pancreas was outlined on each slice of the fat-suppressed T2-weighted image by an experienced radiologist (M.A.H.). Repeat scans of the same individual were blinded and presented to the radiologist non-consecutively to minimize bias. Pancreas outlining was performed using the Medical Image Processing, Analysis, and Visualization (MIPAV) application (https://mipav.cit. nih.gov/). The T1-weighted image was consulted to guide delineation of the pancreas border. Pancreas volume was calculated by multiplying the sum of the pancreas area on each slice by the distance between slices [25]. For human imaging, the pancreas volume was divided by the individual's weight to yield the pancreas volume index (PVI) [5]. The surface area to volume ratio was calculated by summing the perimeter of each pancreas slice, multiplying by the distances between slices to yield surface area, and dividing by the pancreas volume. All image analysis was performed in MATLAB (The MathWorks, Natick, Massachusetts, R2019A).

ADC maps were calculated from the diffusion-weighted images acquired with $b$ values of 200 and 800 s/mm$^2$. ADC values were computed for each voxel by fitting the signal intensities to Eq 1:

$$SI_{(b\_high)} = SI_{(b\_low)} \times e^{-ADC \times (b\_high - b\_low)} \tag{1}$$

where $SI_{(b\_high)}$ and $SI_{(b\_low)}$ are the signal intensities at b values of 800 and 200 s/mm$^2$, respectively. T1 mapping was performed using a variable flip angle (VFA) technique, which is rapid, but prone to inaccuracies due to the $B_1$ inhomogeneities commonly seen at higher fields [26]. To mitigate this issue, we employed a $B_1$ correction map that quantifies the difference between the prescribed and actual flip angles at each voxel. $B_1$-corrected $T_1$ values were calculated for each voxel by fitting the signal intensity, $S$, data to Eq [2]:

$$S = S_0 \bullet \left[ \frac{\sin(f \bullet \alpha) \bullet \left(1 - \exp\left(-\frac{TR}{T_1}\right)\right)}{1 - \left(\exp\left(-\frac{TR}{T_1}\right) \bullet \cos(f \bullet \alpha)\right)} \right], \tag{2}$$

where $S_0$ is a constant related to scanner gain and proton density, $\alpha$ is the prescribed flip angle, and $f$ is the flip angle correction factor that accounts for inhomogeneity in $B_1$. We have assumed $TE << T_2^*$. MTR maps were generated from the gradient echo images performed with and without a magnetization transfer saturation pulse for each voxel using Eq 3:

$$MTR = \frac{SI_{MT\_off} - SI_{MT\_on}}{SI_{MT\_off}} \tag{3}$$

where $SI_{MT\_on}$ and $SI_{MT\_off}$ are the signal intensities with and without the MT saturation pulse, respectively [27] Fat fraction was calculated from 6-point Dixon data using vendor product software on the MRI scanner. For each multiparametric MRI map (ADC, T1, MTR, fat fraction), the pancreas volume outlined by the radiologist was co-registered and re-gridded to the image resolution of each parametric map to generate a whole-pancreas region of interest. Each MRI parameter was then averaged throughout the whole-pancreas region of interest

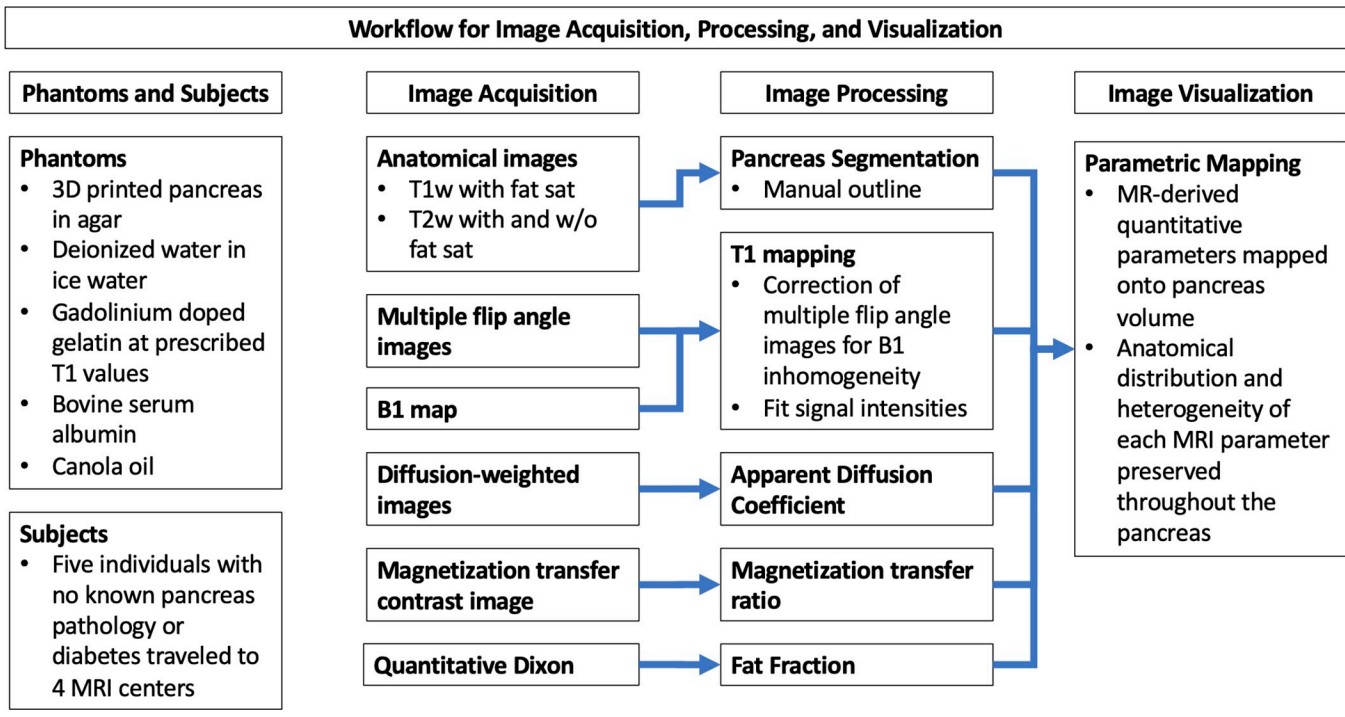

**Fig 2. Flow chart of MRI data acquisition and processing.**

to yield the mean value of each parameter for voxels within the pancreas. For hepatic fat fraction a circular region of interest of area 4 cm$^2$ was placed on a single slice of the right lobe of the liver while carefully avoiding large vessels and bile ducts. A flowchart summarizing data acquisition and image processing is presented in Fig 2.

## Statistical analysis

The reproducibility of each radiomic measure was quantified for phantoms across five MRI scanners and for healthy volunteers across four MRI scanners. For phantom studies, the accuracy of volume and T1 measures were calculated as percent difference between the reference and measured value. For measures without reference values, the mean coefficient of variation across MRI scanners was calculated. Friedman's test was used to assess the difference in each measure across MRI scanners, with post-hoc Wilcoxon rank sum testing to assess differences between each pair of scanners. The difference between scans performed on different individuals on the same scanner (inter-individual variation) was compared with the difference between scans on the same individual using different scanners (inter-assay variation). Power analysis was performed by calculating mean and standard deviation of both inter-individual and inter-assay variation for each MRI measure and calculating the sample size required to achieve a prescribed statistical power. To put our findings in perspective, we also performed power analysis on similar measurements performed on control individuals at a single site (n = 79), an extension of work previously reported [5]. A significance level of $p \leq 0.05$ was used for all statistical tests.

## Results

To standardize and harmonize imaging across the five centers with different MRI technology and software, synthetic imaging phantoms with known MRI properties were created and

shipped to each site for imaging (Fig 1A). Pancreas phantom volume measurements at each of the five sites were similar to the true volume, with a maximum and average difference of 2% from the true value (Table 2). The average difference between T1 measurements at each site and the true values was 6%, with a maximum difference of 13% (Table 2). Of note, one site (Melbourne) had the lowest T1 measurement for each phantom, suggesting there may be bias in T1 mapping on this scanner. The ADC, MTR, and fat fraction measurements were compared across sites, resulting in coefficients of variation of 4.3%, 8.9%, and 0.4%, respectively, across the five sites. We did not detect any temporal changes in these phantoms over the course of these studies. These reproducibility measurements indicate that the acquisition and processing protocol was harmonized across the five sites.

To quantify the reproducibility of quantitative MRI techniques in the pancreas, we sent five volunteers to each of the four MAP-T1D sites in the US (Fig 1B). Multiparametric MRI measurements of the pancreas from these five volunteers are shown in Table 2 and graphically in Fig 3. The average coefficient of variation across five individuals in pancreas volume and PVI were similar at 9.5% and 9.8%, respectively. The ratio of pancreas surface area to volume displayed the lowest coefficient of variation at 5.3%. For the voxel-wise MRI measures, the coefficient of variation was 8.4% for ADC measurements, 9.5% for T1, 39.3% for MTR, 26.5% for pancreatic fat fraction, and 30.2% for hepatic fat fraction. We did not detect a statistically significant difference in any measure across the MRI scanners or between any pair of scanners. For each MRI measure, the difference in measurements between individuals (inter-individual variation) was greater than the difference in measurements made on the same individual on different scanners (inter-assay variation). Dot plots in Fig 3 indicate the distribution of PVI, ADC, and surface area to volume ratio calculated for 79 control subjects at a single site, an extension of our previously published study [5].

Power analysis was performed to estimate the minimum number of subjects required to power clinical trials using MRI measures. Table 3 displays the number of subjects required to detect both differences between two independent samples (e.g., controls versus individuals with diabetes) as well as longitudinal differences in the same individual. Sample sizes are provided to detect 5%, 10%, or 20% changes in each MRI measure at 80% and 90% power. As there was greater inter-individual variation than inter-assay variation for each measure, detecting changes in a single individual, as in a longitudinal study, requires smaller sample sizes. The size measures (volume and PVI) require similar sample sizes to detect changes in an individual, but PVI has added power to detect differences between groups as it accounts for the correlation between body size and pancreas size. The surface area to volume ratio had the highest power for discriminating two groups or detecting changes in an individual. Of the voxel-based measures, ADC and T1 measurements have similar statistical power to pancreas size measures. Pancreatic and hepatic fat fraction have moderate ability to detect changes in an individual, but large inter-individual differences in fat content across individuals limits the ability to detect differences in fat fraction between groups. MTR requires large sample sizes to detect changes in the pancreas. To provide context to these multisite results, we performed derived power measures/indices using a previously acquired dataset of control individuals (n = 79) who underwent longitudinal pancreas MRI on a single MRI scanner [5]. As shown in Table 4, this cohort also displayed greater inter-individual variation than inter-assay variation for each measure and thus was more highly powered to detect changes in an individual than between groups. Sample sizes for detecting changes in an individual were similar for the single site and multisite cohorts, but larger for comparing independent groups in the single site study than the multisite study. The lone exception to this was MTR, which was performed during a breath hold in the single site study which had higher reproducibility.

**Table 2. MRI reproducibility results.**

| Phantom | MRI Measure | MRI Scanner Location | | | | |
|---|---|---|---|---|---|---|
| | | Nashville | Austin | Denver | Chicago | Melbourne |
| Ref. Value: 89.0 | Volume [ml] | 90.8 | 87.4 | 91 | 91.2 | 89.5 |
| | MTR | 0.383 | 0.378 | 0.384 | 0.317 | 0.326 |
| | ADC [mm$^2$/s] | 0.0014 | 0.0014 | 0.0013 | 0.0013 | 0.0015 |
| Ref. Value: 500 | T1 [ms] (vial 1) | 468 | 500 | 494 | 454 | 447 |
| Ref. Value: 1000 | T1 [ms] (vial 2) | 948 | 1038 | 1020 | 1141 | 897 |
| Ref. Value: 1250 | T1 [ms] (vial 3) | 1284 | 1250 | 1256 | 1375 | 1092 |
| Ref. Value: 1500 | T1 [ms] (vial 4) | 1570 | 1550 | 1600 | 1625 | 1354 |
| | Fat Fraction | 96.7 | 95.9 | N/A | 96.2 | 95.6 |
| **Volunteer 1 Pancreas** | | | | | | |
| | Volume [ml] | 81.297 | 59.616 | 60.768 | 67.914 | |
| | PVI [ml/kg] | 1.0862 | 0.79655 | 0.77619 | 0.90742 | |
| | MTR | 0.37469 | 0.44421 | 0.44516 | 0.24861 | |
| | ADC [mm$^2$/s] | 0.0011888 | 0.0010299 | 0.0011715 | 0.0012272 | |
| | T1 [ms] | 958.61 | 890.77 | 866.19 | 1043.3 | |
| | Pancreatic Fat Fraction | 0.1472 | 0.13584 | N/A | 0.18757 | |
| | Hepatic Fat Fraction | 0.0422 | 0.0296 | N/A | 0.0312 | |
| | Surface area/volume [cm$^{-1}$] | 0.1017 | 0.10107 | 0.10482 | 0.095063 | |
| **Volunteer 2 Pancreas** | | | | | | |
| | Volume [ml] | 73.602 | 74.214 | 87.93 | 73.134 | |
| | PVI [ml/kg] | 1.081765 | 1.09808 | 1.294076 | 1.07488 | |
| | MTR | 0.41664 | 0.30807 | 0.31042 | 0.055648 | |
| | ADC [mm$^2$/s] | 0.0012611 | 0.0012671 | 0.0014773 | 0.0014198 | |
| | T1 [ms] | 1108.1 | 1236.4 | 1311.6 | 1055.5 | |
| | Pancreatic Fat Fraction | 0.020894 | N/A | N/A | 0.060248 | |
| | Hepatic Fat Fraction | 0.0465 | N/A | N/A | 0.024 | |
| | Surface area/volume [cm$^{-1}$] | 0.1118 | 0.10107 | 0.10595 | 0.11043 | |
| **Volunteer 3 Pancreas** | | | | | | |
| | Volume [ml] | 112.75 | 122.29 | 127.75 | 112.9 | |
| | PVI [ml/kg] | 1.1893 | 1.29617 | 1.30389 | 1.1852 | |
| | MTR | 0.40765 | 0.28954 | 0.34684 | N/A | |
| | ADC [mm$^2$/s] | 0.0012716 | 0.001267 | 0.0011184 | 0.0010943 | |
| | T1 [ms] | 1168.1 | 874.06 | 818.81 | 989.93 | |
| | Pancreatic Fat Fraction | 0.13 | 0.13369 | N/A | 0.12877 | |
| | Hepatic Fat Fraction | 0.0394 | 0.0293 | N/A | 0.0185 | |
| | Surface area/volume [cm$^{-1}$] | 0.087008 | 0.086092 | 0.075841 | 0.095138 | |
| **Volunteer 4 Pancreas** | | | | | | |
| | Volume [ml] | 57.177 | 60.381 | 63.351 | 70.839 | |
| | PVI [ml/kg] | 0.7781 | 0.80677 | 0.84645 | 0.97 | |
| | MTR | 0.249 | 0.32639 | 0.3998 | 0.085408 | |
| | ADC [mm$^2$/s] | 0.0013227 | 0.0014217 | 0.0011917 | 0.0011398 | |
| | T1 [ms] | 901.82 | 897.35 | 868.17 | 918.05 | |
| | Pancreatic Fat Fraction | 0.077119 | 0.078579 | N/A | 0.13843 | |
| | Hepatic Fat Fraction | 0.0298 | 0.0163 | N/A | 0.0269 | |
| | Surface area/volume [cm$^{-1}$] | 0.11663 | 0.11427 | 0.11303 | 0.11338 | |
| **Volunteer 5 Pancreas** | | | | | | |
| | Volume [ml] | 96.57 | 116.41 | 103.18 | 100.39 | |

*(Continued)*

**Table 2.** (Continued)

| Phantom | MRI Measure | MRI Scanner Location | | | | |
|---|---|---|---|---|---|---|
| | | Nashville | Austin | Denver | Chicago | Melbourne |
| | PVI [ml/kg] | 0.9721 | 1.19367 | 1.04345 | 1.0294 | |
| | MTR | 0.39885 | 0.40804 | 0.39628 | 0.089641 | |
| | ADC [mm$^2$/s] | 0.0010012 | 0.00094109 | 0.0008776 | 0.0010638 | |
| | T1 [ms] | 676.91 | 839.37 | 830.25 | 707.06 | |
| | Pancreatic Fat Fraction | 0.23798 | 0.2386 | N/A | 0.20345 | |
| | Hepatic Fat Fraction | 0.1381 | 0.1111 | N/A | 0.1686 | |
| | Surface area/volume [cm$^{-1}$] | 0.075479 | 0.083718 | 0.070601 | 0.081018 | |

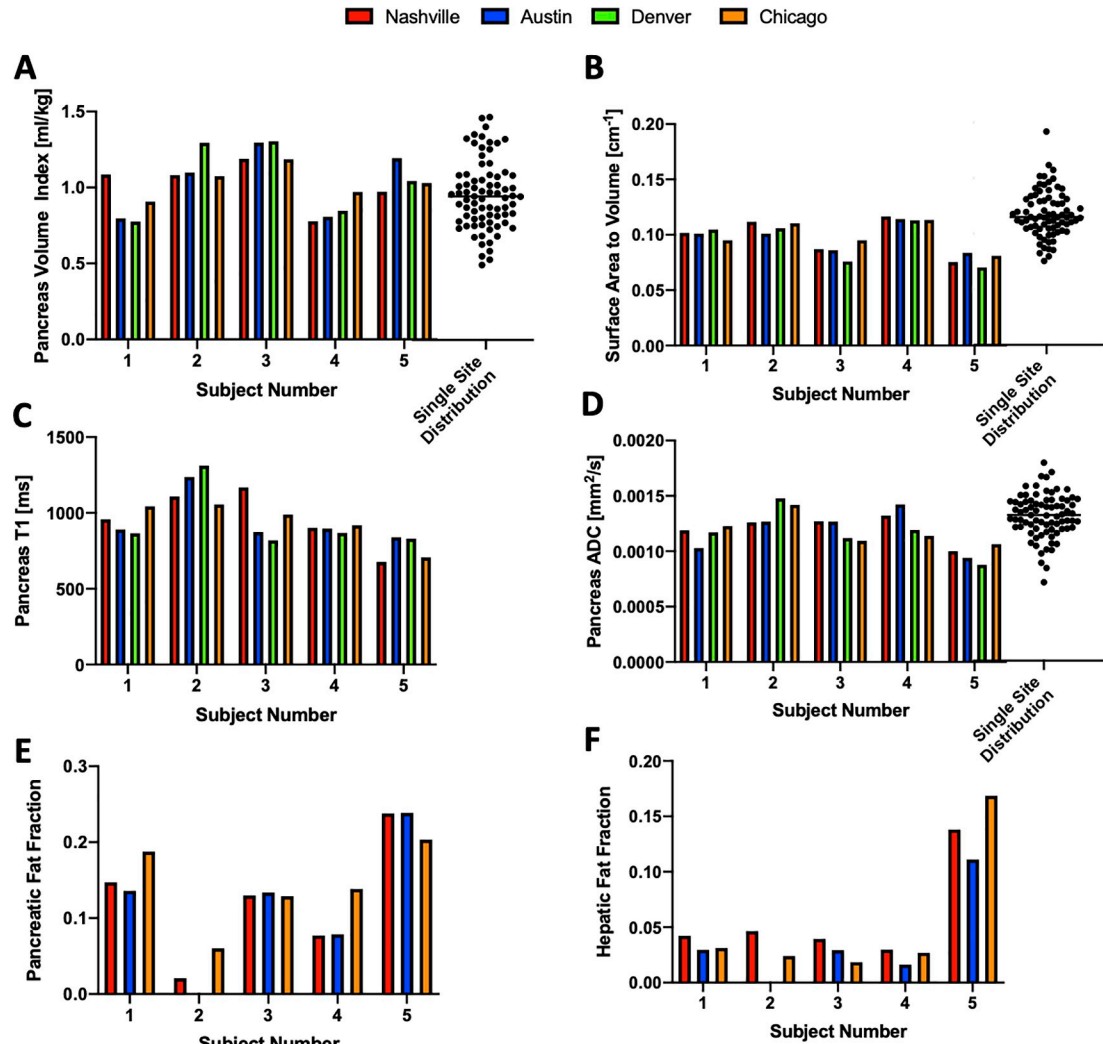

**Fig 3. Quantitative MRI measures for 5 individuals scanned on four different MRI centers.** Values for each MRI measurement of the pancreas are displayed for: A) pancreas volume index (PVI), B) surface area to volume ratio, C) longitudinal relaxation time (T1), D) apparent diffusion coefficient (ADC), E) pancreatic fat fraction, and F) hepatic fat fraction. Note that fat fraction was not measured in Denver due to a lack of the requisite software. For the graphs of PVI, surface area to volume ratio, and ADC, the distribution of values calculated in healthy volunteers at a single site study (updated from a previously published study [5]) is indicated by dot plot in panels A, B, D.

**Table 3. Projected number of study participants required for future clinical trial (multisite).**

| | Two Independent Groups | | | | | |
|---|---|---|---|---|---|---|
| **MRI Measure** | 80% Power | | | 90% Power | | |
| | 5% Difference | 10% Difference | 20% Difference | 5% Difference | 10% Difference | 20% Difference |
| Volume [ml] | 930 | 234 | 60 | 1244 | 312 | 80 |
| PVI [ml/kg] | 362 | 92 | 24 | 484 | 122 | 32 |
| ADC [mm$^2$/s] | 504 | 128 | 34 | 674 | 170 | 44 |
| T1 [ms] | 380 | 96 | 26 | 508 | 128 | 34 |
| MTR | 2244 | 562 | 142 | 3004 | 752 | 190 |
| Pancreatic Fat Fraction | 1630 | 408 | 104 | 2180 | 546 | 138 |
| Hepatic Fat Fraction | 9434 | 2360 | 850 | 12,630 | 3158 | 1138 |
| Surface area/volume [cm$^{-1}$] | 284 | 72 | 20 | 380 | 96 | 26 |
| | **Within Subject Variation** | | | | | |
| **MRI Measure** | 80% Power | | | 90% Power | | |
| | 5% Difference | 10% Difference | 20% Difference | 5% Difference | 10% Difference | 20% Difference |
| Volume [ml] | 28 | 9 | 4 | 36 | 11 | 4 |
| PVI [ml/kg] | 30 | 9 | 4 | 39 | 11 | 5 |
| ADC [mm$^2$/s] | 33 | 10 | 4 | 44 | 13 | 5 |
| T1 [ms] | 31 | 9 | 4 | 41 | 12 | 5 |
| MTR | 589 | 149 | 39 | 787 | 198 | 51 |
| Pancreatic Fat Fraction | 64 | 17 | 6 | 84 | 23 | 7 |
| Hepatic Fat Fraction | 149 | 39 | 15 | 199 | 51 | 20 |
| Surface area/volume [cm$^{-1}$] | 10 | 4 | 3 | 13 | 5 | 3 |

To examine the impact of different image acquisition and processing, we performed image processing using both a standardized and non-standardized protocol. To generate the non-standardized image T1 map, B1 field correction was not performed. To simulate non-standardized diffusion processing, we changed the b-values employed to generate ADC maps from 200 and 800 in the standardized protocol to 0 and 800 in the non-standardized protocol. The

**Table 4. Projected number of study participants required for future clinical trial (single site).**

| | Two Independent Groups | | | | | |
|---|---|---|---|---|---|---|
| **MRI Measure** | 80% Power | | | 90% Power | | |
| | 5% Difference | 10% Difference | 20% Difference | 5% Difference | 10% Difference | 20% Difference |
| Volume [ml] | 2142 | 538 | 136 | 2868 | 718 | 182 |
| PVI [ml/kg] | 768 | 198 | 50 | 1026 | 264 | 68 |
| ADC [mm$^2$/s] | 258 | 128 | 16 | 346 | 170 | 20 |
| MTR | 722 | 182 | 46 | 964 | 242 | 60 |
| | **Within Subject Variation** | | | | | |
| **MRI Measure** | 80% Power | | | 90% Power | | |
| | 5% Difference | 10% Difference | 20% Difference | 5% Difference | 10% Difference | 20% Difference |
| Volume [ml] | 40 | 12 | 5 | 53 | 15 | 5 |
| PVI [ml/kg] | 36 | 11 | 4 | 48 | 14 | 5 |
| ADC [mm$^2$/s] | 18 | 10 | 3 | 23 | 13 | 3 |
| MTR | 128 | 16 | 6 | 170 | 21 | 7 |

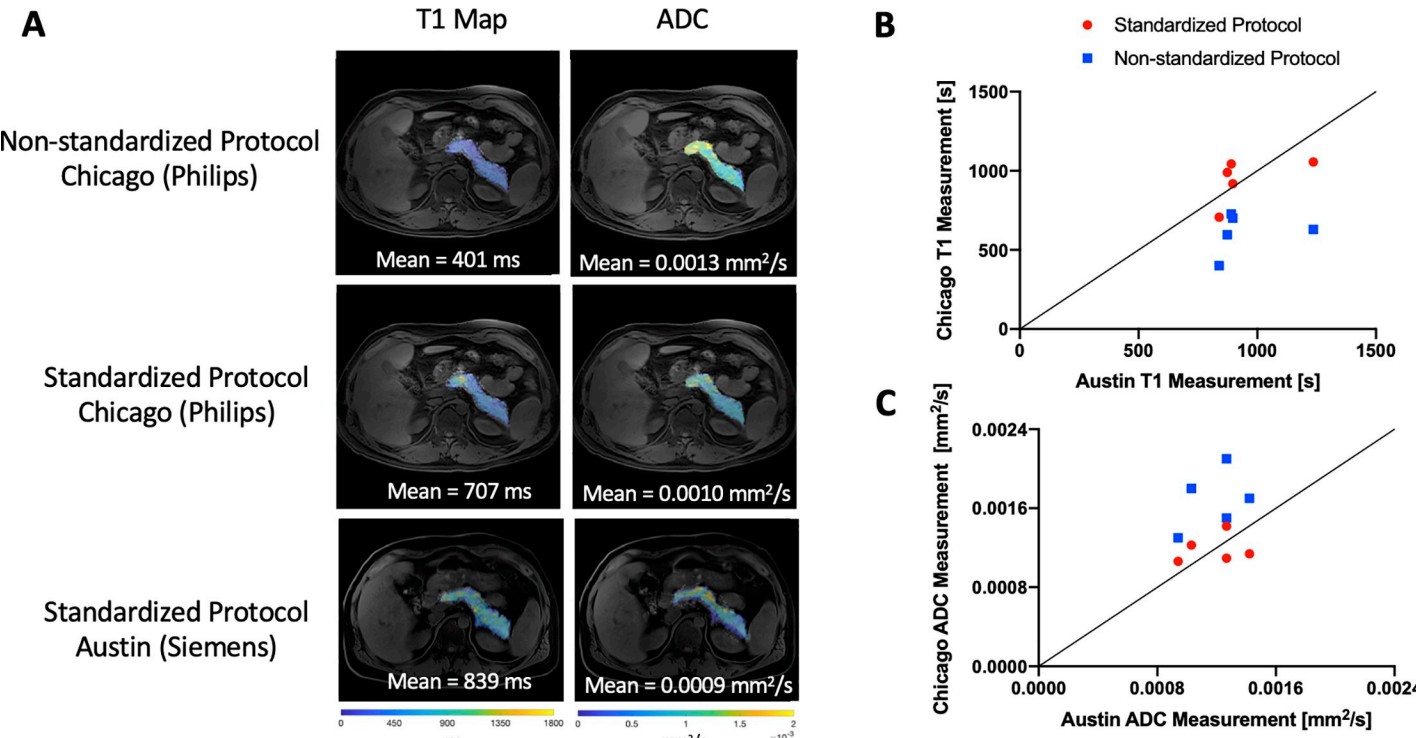

**Fig 4. Representative difference in quantitative MRI measurements induced by use of different image processing between sites.** A) Representative maps of T1 relaxation time (left column) and ADC (right column) displayed in pseudo color over a T1-weighted image. The top row displays images acquired in Chicago and processed using a non-standardized image processing protocol, demonstrating differences in T1 and ADC values from the standardized protocol (middle row). Images acquired and processed using the standardized MAP-T1D protocol in Chicago on a Philips MRI scanner (middle row) and Austin on a Siemens MRI scanner (bottom row) display concordance for T1 and ADC. All sets of parametric maps are scaled identically for visualization. B) Mean pancreatic T1 values are more reproducible between two sites (Chicago and Austin) using the standardized image analysis protocol (red circles), than when using non-standardized image processing (blue squares). The line of identity indicates perfect agreement. C) Mean pancreatic ADC values are more reproducible between two sites (Chicago and Austin) when using the standardized image analysis protocol (red circles), than when using non-standardized image processing (blue squares).

results of this analysis for subjects imaged at two different sites (Austin and Chicago) are shown in Fig 4. Representative non-standardized T1 and ADC maps are shown in the top row of Fig 4A. In contrast, images acquired and processed using the standardized MAP-T1D protocol at two different sites (Fig 4A, middle and bottom row) display similar values for both T1 and ADC. For each individual, the mean pancreas T1 value calculated using the standardized protocol at two sites was more reproducible than when a non-standardized protocol was employed at one site (Fig 4B). Similarly, the mean pancreatic ADC value calculated using the standardized protocol at two sites displayed better agreement than when a non-standardized protocol was employed (Fig 4C). We calculated the variation induced by using a non-standardized MRI protocol to estimate sample sizes required if MRI protocols are not standardized. A non-standardized MRI protocol with 80% power would require a sample size 22-fold greater to detect a 10% change in ADC and 10-fold greater to detect a 10% change in T1.

## Discussion

This study is the first effort to generate a standardized pancreas MRI protocol for techniques that are being used for studying diabetes. When deployed to five different MRI centers, this standardized protocol demonstrated the reproducibility of quantitative imaging measures using both calibrated phantoms and healthy volunteers. Phantom measurements at five

different sites demonstrated excellent accuracy to known standards and excellent reproducibility across sites. In pancreas scans in humans, MRI measurements of pancreas volume, ADC, T1, fat fraction, and surface area/volume ratio using a standardized protocol were reproducible across different scanner hardware and software. The harmonized image acquisition and processing tools developed in this study have been made available to anyone interested and can be deployed in multisite clinical trials incorporating pancreas imaging.

Previous MRI studies of the pancreas of individuals with diabetes have led to conflicting results, which likely stem, in part, from disparities in image acquisition and processing. For example, a meta-analysis of pancreatic volume and fat content found high heterogeneity between studies [28]. However, we found that MRI of the pancreas performed at a single site is repeatable [29]. The choice of image acquisition and processing parameters can influence a calculated measure, as previously demonstrated in pancreatic measurements of fat fraction [30], T1 [31] and ADC [32]. Given this dependence on imaging parameters, efforts to standardize MRI protocols for different diseases and anatomies are underway. Neuroimaging studies have been at the forefront of this standardization and have shown that standardized protocols greatly improve the statistical power of multisite studies [33]. Despite evidence promising new insights from pancreas imaging related to diabetes, standardized pancreatic imaging protocols are underdeveloped. While a standardized MRI protocol for imaging of pancreatitis has been proposed [34], it has not been rigorously evaluated using phantoms with known MRI properties or scans on the same individuals across sites. Thus, the current report provides a new approach to integrating pancreas MRI into multisite clinical trials in diabetes and pancreatitis.

We found high accuracy in volume and T1 measurements using phantoms standardized to known values. Of note, an ice water phantom similar to the one used in this study has previously calculated an ADC of 0.0011 mm$^2$/s [23]. However, results from this work, our previous studies [22], and others [35] have found higher ADC values in ice water phantoms, possibly due to phantom heating or positioning of the phantom at the edge of the field of view where gradient nonlinearity is present. Therefore reproducibility, but not accuracy, of ADC was calculated in this study. While a systemic bias internal to the phantom is evident, the high reproducibility of the measured ADC values across five different scanners is reassuring.

For in vivo reproducibility studies, MRI measures displayed marked differences among individuals with no known pancreas pathology. For example, one individual (denoted as Volunteer 5 in Table 2) had increased fat content in both the liver and pancreas. An association between fat accumulation the liver and pancreas has been previously reported [36]. It is known that pancreatic fat content [37] and diffusion measurements are influenced by age and sex [38]. Importantly, the volunteers scanned in this study were similarly aged, and we did not detect a difference in the female subject. The natural inter-individual variability potentially complicates the use of pancreatic imaging in multisite trials, but our power estimates show that using a standardized protocol could decrease the required number of subjects to be enrolled by ten-fold or more. Importantly, the inter-individual differences in MRI measures were preserved across scanners, demonstrating that MRI measures (other than MTR) yield reproducible assays of pancreas size and structure. Measurements of pancreatic MTR were not reproducible in volunteers scanned at multiple sites. This observed variation is likely due to respiratory motion artifacts from this free-breathing acquisition, as phantom MTR values were reproducible and there was higher reproducibility in single-site data where a breath-hold was employed during acquisition [5]. Importantly, this larger cohort of longitudinal scans performed at a single site also displayed higher power for detecting intra-individual changes than comparing across groups. The single site study displayed higher inter-individual variation, likely due to the longitudinal study design and increased heterogeneity in cohort

demographics. However, inter-assay variation was similar in the single site and multisite study, suggesting that imaging on multiple scanners did not add variation in any MRI parameter when a standardized protocol was used. In summary, longitudinal monitoring maximizes statistical power and may have important applications in diabetes prediction, staging, and prognosis, as has previously been shown in type 1 diabetes [5].

This study is subject to a number of limitations. MRI scanners from both Philips and Siemens were represented in the study sites, but GE scanners were not part of the current study. Additionally, all scans were performed at 3T. Several quantitative MRI parameters can be influenced by field strength, and thus translation of this protocol to 1.5T field strength will likely require adjustment of imaging parameters. Finally, a larger volunteer cohort would allow for a more precise evaluation of variability. Sending the same individual to each study site for scanning is both expensive and technically difficult, and thus, it is not commonly performed. Given the complexity of navigating volunteer travel and scheduling, we were limited to five individuals scanned at four different sites but showed good agreement in MRI measurements across a wide variety of MRI hardware and software. An additional limitation of this study is the use of a single reader to outline all pancreas images in this study. The use of multiple readers may lead to variation in MRI measurements of the pancreas [29]. Magnetic resonance elastography is another promising technique for assessing the pancreas, but as it requires specialized hardware and software not available at our study sites it was not included in this study. However, a previous study examining the reproducibility of magnetic resonance elastography in healthy controls indicates that it has similar reproducibility to the T1 and ADC measurements in this study [39].

This study demonstrates that, when carefully controlled and standardized, quantitative MRI of the pancreas is highly reproducible across different MRI hardware and software at different geographic locations. Pancreas MRI can now be incorporated into multisite clinical trials of diabetes and other pancreatic diseases. In order to standardize acquisition and processing of MRI studies of the pancreas, we have made our image acquisition protocols and image processing code freely available for any users using Github (https://github.com/jvirostko/MAPT1D). It is our hope that this protocol will be adapted and modified by other groups in the diabetes and imaging community performing MRI of the pancreas. Efforts in employing a common standardized protocol will improve data quality and reporting and facilitate comparison of results across sites and, ultimately, multisite clinical trials. Furthermore, use of a common image acquisition standard will empower future applications of machine learning to studies of pancreas MRI.

## Acknowledgments

The University of Chicago would like to thank the Kovler Diabetes Center research team including Mariko Pusinelli, Rabia Ali, and Cristy Miles, for their invaluable help with all aspects of study coordination. We sincerely thank all research participants who took part in this in this study.

## Author Contributions

**Conceptualization:** John Virostko, Siri Atma W. Greeley, Daniel J. Moore, Alvin C. Powers.

**Data curation:** John Virostko, Hakmook Kang, Liping Du.

**Formal analysis:** John Virostko, Richard C. Craddock, Taylor M. Triolo, Melissa A. Hilmes, Hakmook Kang, Liping Du, Jordan J. Wright, Jeffrey H. Maki, Daniel J. Moore, Alvin C. Powers.

**Funding acquisition:** John Virostko, Alvin C. Powers.

**Investigation:** John Virostko, Jonathan M. Williams, Taylor M. Triolo, Mara Kinney, Milica Medved, Michaela Waibel, Thomas W. H. Kay, Helen E. Thomas, Siri Atma W. Greeley, Andrea K. Steck, Daniel J. Moore, Alvin C. Powers.

**Methodology:** John Virostko.

**Software:** John Virostko, Richard C. Craddock.

**Writing – original draft:** John Virostko, Daniel J. Moore, Alvin C. Powers.

**Writing – review & editing:** John Virostko, Richard C. Craddock, Jonathan M. Williams, Taylor M. Triolo, Melissa A. Hilmes, Hakmook Kang, Liping Du, Jordan J. Wright, Mara Kinney, Jeffrey H. Maki, Milica Medved, Michaela Waibel, Thomas W. H. Kay, Helen E. Thomas, Siri Atma W. Greeley, Andrea K. Steck, Daniel J. Moore, Alvin C. Powers.

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
