## [Decision Letter · Decision Letter 0]

2 Jun 2021

PONE-D-21-07115

Development of a Standardized MRI Protocol for Pancreas Assessment in Humans

PLOS ONE

Dear Dr. Virostko,

Thank you for submitting your manuscript to PLOS ONE. After careful consideration, we feel that it has merit but does not fully meet PLOS ONE’s publication criteria as it currently stands. Therefore, we invite you to submit a revised version of the manuscript that addresses the points raised during the review process.

We look forward to receiving your revised manuscript.

Kind regards,

Mingwu Jin, Ph.D.

Academic Editor

PLOS ONE

Additional Editor Comments:

The manuscript represents an important and well-organized work for standardizing the MRI protocol for pancreas assessment. Addressing the issues raised by the reviewers could further strengthen its impact.

Journal Requirements:

3. We note that Figure 1 in your submission contain map images which may be copyrighted. All PLOS content is published under the Creative Commons Attribution License (CC BY 4.0), which means that the manuscript, images, and Supporting Information files will be freely available online, and any third party is permitted to access, download, copy, distribute, and use these materials in any way, even commercially, with proper attribution. For these reasons, we cannot publish previously copyrighted maps or satellite images created using proprietary data, such as Google software (Google Maps, Street View, and Earth). For more information, see our copyright guidelines: http://journals.plos.org/plosone/s/licenses-and-copyright.

3.1.    You may seek permission from the original copyright holder of Figure 1 to publish the content specifically under the CC BY 4.0 license. 

3.2.    If you are unable to obtain permission from the original copyright holder to publish these figures under the CC BY 4.0 license or if the copyright holder’s requirements are incompatible with the CC BY 4.0 license, please either i) remove the figure or ii) supply a replacement figure that complies with the CC BY 4.0 license. Please check copyright information on all replacement figures and update the figure caption with source information. If applicable, please specify in the figure caption text when a figure is similar but not identical to the original image and is therefore for illustrative purposes only.

Reviewers' comments:

Reviewer's Responses to Questions

**Comments to the Author**

1. Is the manuscript technically sound, and do the data support the conclusions?

Reviewer #1: Yes

Reviewer #2: Yes

2. Has the statistical analysis been performed appropriately and rigorously? 

Reviewer #1: Yes

Reviewer #2: Yes

3. Have the authors made all data underlying the findings in their manuscript fully available?

Reviewer #1: Yes

Reviewer #2: Yes

4. Is the manuscript presented in an intelligible fashion and written in standard English?

Reviewer #1: Yes

Reviewer #2: Yes

5. Review Comments to the Author

Reviewer #1: The authors proposed a non-contrast quantitative MR imaging protocol for pancreas, including DWI, T1, MTR and PDFF (liver & pancreas). The purpose of this protocol is to provide quantitative matrix for diagnosing and monitoring pancreatic abnormalities caused by chronic or acute pancreatitis, NAFLD, diabetes, and pancreatic cancer. Since obesity and NAFLD are risk factors in the development of diabetes, evaluation of adipose tissue and the liver would strengthen this protocol. In the quantitative imaging field, MRE is also an important method to detect tissue abnormalities (inflammation and fibrosis) via mechanical response against applied vibration. It has been well-established that liver stiffness can be a reliable alternative to biopsy for assessing hepatic fibrosis and providing prognostic value as well. There are many publications about pancreatic MRE from Dr. Yu Shi in this field. The author could make a complete imaging characterization by adding this imaging method.

1. Methods: what’s the time range for phantom scan between locations? According to Ref 23, there was little change in a longitudinal study of 25days for the ice water phantom, but no mention of longer time. In Ref 24, they used 5% and 10% BSA, no mention of 15%, or any dynamic changes. Will the reference value change during time? If not, any other reference?

2. Methods: how big are the ROIs for calculating fat fraction, pancreas and liver, respectively? And where are the ROIs located, since the fat fraction varies much between locations.

3. Results: the hepatic fat fraction of Volunteers 5, is over 5%, which is diagnosed as fatty liver. In that case, it might influence the measurements of pancreas.

4. Discussion: the measurements of fat fraction are all missing in Denver, is there any reason?

5. Figure 1: the figure and descriptions in manuscript should be consistent. eg, in the manuscript Methods_Phantom Studies, it was described as T1, MTR, and fat fraction, in the figure it showed “T1/MTR/Fat” in the top row , but the bottom row indicate Canola oil, BSA, and Gadolinium, which are used for fat fraction, MTR, and T1, respectively. Besides, it is also really confusing with three same color pattern and no unit or label at all.

6. Figure 4A, please indicate all the measurements, it is difficult to tell changes from the color.

Reviewer #2: This manuscript presents a standardized MRI protocol for pancreas assessment in humans and define the reproducibility of all kind of measurements, such as pancreas size, shape, ADC, T1 relaxation time, MTR and fat fractions. These quantitative MRI matrixes of the pancreas were performed at multiple sites. Physics phantoms were constructed for protocol standardization including pancreas volume, ADC, T1, MTR, and fat fractions. Also, the same 5 healthy volunteers with no known pancreas pathology or diabetes traveled to multiple sites to standardize the protocols. In addition to these scanning technique standardizations, images of the phantoms and human volunteers were analyzed using the same methodology. The reproducibility was quantified for phantoms across different MRI scanners and for healthy volunteers across four MRI scanners. These techniques are very helpful for standardize the protocols. The authors provided very meaningful data to support the conclusions. The manuscripts were written and structured very well. Because the choice of protocol standardization in multiple vendors and different locations impacts image quality in pancreas MRI studies, this study is a useful topic for radiologists and clinicians to know about. Overall, a very comprehensive pancreas MRI protocol optimizing study that has the potential to serve as a good reference/resource for the readers. There are some concerns listed below:

1. In MRI Scanning Protocol section, list in a table of scanner vendors, model and software version will be helpful for understanding the results.

2. In Table 1, can it include the bandwidth? b vectors for DWI? Regarding to T1 mapping, what does “5 acquisitions: 5 flip angles”? 5 series?

3. In table 2, it seems the quantitative T1 values in via ,2,3,4 in Melbourne is lower than these in other sites? Is it the system bias?

4. Could you please comment on that if we can normalize the data with standardized phantom data for the quantitative measurements in humans/individuals?

5. How many radiologists were involved in for outlining pancreas volume? Were the same radiologists involved in doing the same work for all sites’ studies? If not, any bias?

6. More volunteer’s data will be better.

6. PLOS authors have the option to publish the peer review history of their article (what does this mean?). If published, this will include your full peer review and any attached files.

Reviewer #1: No

Reviewer #2: No

---

## [Author Response · Author response to Decision Letter 0]

25 Jun 2021

June 21, 2021

Dear Dr. Jin:

Thank you for review of our manuscript entitled ‘Development of a Standardized MRI Protocol for Pancreas Assessment in Humans’ for consideration by the journal PLOS ONE.

We have revised our manuscript based upon the constructive criticism of the reviewers and believe that the revised manuscript is greatly improved. The revised manuscript is provided with both ‘tracked changes’ and a ‘clean’ version. We address each reviewer comment below (author response in italics):

Journal Requirements:

Author Response: The manuscript has been reformatted according to PLOS ONE style guidelines. 

Author Response: We would like to alter our Data Availability statement to match that of our submitted manuscript: ‘The datasets generated during and/or analyzed during the current study are available from the corresponding author upon reasonable request. The image acquisition and processing protocols used in these studies are available online: https://github.com/jvirostko/MAPT1D.’ This change has been noted in a revised cover letter accompanying this resubmission. 

3. We note that Figure 1 in your submission contain map images which may be copyrighted… If you are unable to obtain permission from the original copyright holder to publish these figures under the CC BY 4.0 license or if the copyright holder’s requirements are incompatible with the CC BY 4.0 license, please either i) remove the figure or ii) supply a replacement figure that complies with the CC BY 4.0 license.

Author Response: The map in Figure 1B has been replaced with a US map generated by our group.

Reviewer #1: 

1. Since obesity and NAFLD are risk factors in the development of diabetes, evaluation of adipose tissue and the liver would strengthen this protocol.

Author Response: We agree with the reviewer and respectfully note that we have included liver fat analysis in the current manuscript. However, in order to focus on pancreas imaging, which is currently less developed than liver imaging, we have chosen to limit the studies of liver imaging in this work. Liver and adipose tissue reproducibility will be analyzed in future studies. 

2. It has been well-established that liver stiffness can be a reliable alternative to biopsy for assessing hepatic fibrosis and providing prognostic value as well. There are many publications about pancreatic MRE from Dr. Yu Shi in this field. The author could make a complete imaging characterization by adding this imaging method.

Author Response: We have added a citation to Dr. Shi’s work and MRE to the discussion section: ‘Magnetic resonance elastography is another promising technique for assessing the pancreas, but as it requires specialized hardware and software not available at our study sites it was not included in this study. However, a previous study examining the reproducibility of magnetic resonance elastography in healthy controls indicates that it has similar reproducibility to the T1 and ADC measurements in this study (39).’

3. Methods: what’s the time range for phantom scan between locations? According to Ref 23, there was little change in a longitudinal study of 25days for the ice water phantom, but no mention of longer time. In Ref 24, they used 5% and 10% BSA, no mention of 15%, or any dynamic changes. Will the reference value change during time? If not, any other reference?

Author Response: All phantoms were scanned within 60 days of each other, and we did not detect any differences over the timescale of these studies. This has been added to the methods and results section. 

4. Methods: how big are the ROIs for calculating fat fraction, pancreas and liver, respectively? And where are the ROIs located, since the fat fraction varies much between locations.

Author Response: We apologize for the lack of clarity and have edited the Methods section to provide these details as follows: ‘For each multiparametric MRI map (ADC, T1, MTR, fat fraction), the pancreas volume outlined by the radiologist was co-registered and re-gridded to the image resolution of each parametric map to generate a whole-pancreas region of interest. Each MRI parameter was then averaged throughout the whole-pancreas region of interest to yield the mean value of each parameter for voxels within the pancreas. For hepatic fat fraction a circular region of interest of area 4 cm2 was placed on a single slice of the right lobe of the liver while carefully avoiding large vessels and bile ducts.’

5. Results: the hepatic fat fraction of Volunteers 5, is over 5%, which is diagnosed as fatty liver. In that case, it might influence the measurements of pancreas.

Author Response: We agree with the reviewer that this is an important finding and have added the association between liver and pancreas fat to the discussion: ‘. For example, one individual (denoted as Volunteer 5 in Table 2) had increased fat content in both the liver and pancreas. An association between fat accumulation the liver and pancreas has been previously reported (36).’

6. Discussion: the measurements of fat fraction are all missing in Denver, is there any reason?

Author Response: The scanner in Denver did not have software required for acquiring fat fraction maps as noted in the Methods: ‘A 3D quantitative 6-point Dixon acquisition was collected at sites which had the requisite software on their scanner (Austin, Nashville, Chicago, Melbourne)’. This is also noted in the legend for Figure 3: ‘Note that fat fraction was not measured in Denver due to a lack of the requisite software.’

7. Figure 1: the figure and descriptions in manuscript should be consistent. eg, in the manuscript Methods_Phantom Studies, it was described as T1, MTR, and fat fraction, in the figure it showed “T1/MTR/Fat” in the top row , but the bottom row indicate Canola oil, BSA, and Gadolinium, which are used for fat fraction, MTR, and T1, respectively. Besides, it is also really confusing with three same color pattern and no unit or label at all.

Author Response: We apologize for this inconsistency. We have relabeled the panels in Figure 1 to remain consistent and added units to each colorbar.

8. Figure 4A, please indicate all the measurements, it is difficult to tell changes from the color.

Author Response: Mean pancreas values for T1 and ADC have been added to each panel of Figure 4A to aid interpretation of these figures. 

Reviewer #2: 

1. In MRI Scanning Protocol section, list in a table of scanner vendors, model and software version will be helpful for understanding the results.

Author Response: We thank the reviewer for catching our omission of scanner software. We have added it along with the description of the scanner vendor and models to the first paragraph of the methods section: ‘Two centers (Austin and Denver) were equipped with 3T Skyra scanners (Siemens, Erlangen, Germany). Scans performed in Melbourne were acquired using a 3T Prisma scanner (Siemens, Erlangen, Germany). Two centers (Nashville and Chicago) acquired images on 3T Ingenia scanners (Philips, Best, Netherlands). All Siemens scanners employed VE11C software. Of the Philips scanners, Vanderbilt employed R5.5.0.1 while Chicago employed R.5.6.1.’ As there are a number of repeated values (e.g., all Siemens sites use the same software, we believe that another table is not needed to present these parameters. 

2. In Table 1, can it include the bandwidth? b vectors for DWI? Regarding to T1 mapping, what does “5 acquisitions: 5 flip angles”? 5 series?

Author Response: The b-values for DWI are provided in the ‘Miscellaneous’ row under the ‘DWI’ column in Table 1. We apologize for the confusion in the T1 mapping column. We have rewritten this to ‘5 flip angles’ to limit reader confusion. Bandwidth was not standardized across all scanners due to differences in sequence parameters and thus is not included in Table 1. 

3. In table 2, it seems the quantitative T1 values in via ,2,3,4 in Melbourne is lower than these in other sites? Is it the system bias?

Author Response: We have added the possibility of bias to the results section: ‘Of note, one site (Melbourne) had the lowest T1 measurement for each phantom, suggesting there may be bias in T1 mapping on this scanner.’

4. Could you please comment on that if we can normalize the data with standardized phantom data for the quantitative measurements in humans/individuals?

Author Response: This is a really interesting idea. Unfortunately, preliminary standardization based on phantom results do not improve reproducibility of human scans. This may stem in part from the fact that the phantom scans were largely reproducible. Unfortunately, human scans were not performed in Melbourne due to travel cost concerns. As the reviewer astutely previously noted, there may be bias in T1 measurement in Melbourne which may be improved by standardizing to phantom bias. 

5. How many radiologists were involved in for outlining pancreas volume? Were the same radiologists involved in doing the same work for all sites’ studies? If not, any bias?

Author Response: A single radiologist outlined all pancreas images as described in the Methods section: ‘The pancreas was outlined on each slice of the fat-suppressed T2-weighted image by an experienced radiologist (M.A.H.).’ We have added this use of a single reader as a limitation of this study to the Discussion section: ‘An additional limitation of this study is the use of a single reader to outline all pancreas images in this study. Multiple readers may lead to variation in the MRI measurements performed in this study (29).’

6. More volunteer’s data will be better.

Author Response: We agree with the reviewer that more subjects would increase the power of this work. Unfortunately, the cost of flying each volunteer to three separate locations limited the current study to 5 volunteers each scanned at 4 locations. We note this limitation in the Discussion section of this manuscript: ‘Finally, a larger volunteer cohort would allow for a more precise evaluation of variability. Sending the same individual to each study site for scanning is both expensive and technically difficult, and thus, it is not commonly performed. Given the complexity of navigating volunteer travel and scheduling, we were limited to five individuals scanned at four different sites but showed good agreement in MRI measurements across a wide variety of MRI hardware and software.’ We also note that our previous study of breast MRI reproducibility included only 3 volunteers due to the logistical difficulty of these multisite MRI studies (J Magn Reson Imaging. : 10.1002/jmri.26011.)

Thank you for your input into this work. Please address all correspondence concerning this manuscript to me at jack.virostko@austin.utexas.edu.

Sincerely,

John (Jack) Virostko 

512-495-5126 

jack.virostko@austin.utexas.edu

---

## [Decision Letter · Decision Letter 1]

29 Jul 2021

Development of a Standardized MRI Protocol for Pancreas Assessment in Humans

PONE-D-21-07115R1

Dear Dr. Virostko,

We’re pleased to inform you that your manuscript has been judged scientifically suitable for publication and will be formally accepted for publication once it meets all outstanding technical requirements.

Kind regards,

Mingwu Jin, Ph.D.

Academic Editor

PLOS ONE

Additional Editor Comments (optional):

Please improve Fig. 1 and Fig. 4 suggested by Reviewer 1 for the final version.

Reviewers' comments:

Reviewer's Responses to Questions

**Comments to the Author**

1. If the authors have adequately addressed your comments raised in a previous round of review and you feel that this manuscript is now acceptable for publication, you may indicate that here to bypass the “Comments to the Author” section, enter your conflict of interest statement in the “Confidential to Editor” section, and submit your "Accept" recommendation.

Reviewer #1: All comments have been addressed

Reviewer #2: All comments have been addressed

2. Is the manuscript technically sound, and do the data support the conclusions?

Reviewer #1: Yes

Reviewer #2: (No Response)

3. Has the statistical analysis been performed appropriately and rigorously? 

Reviewer #1: Yes

Reviewer #2: (No Response)

4. Have the authors made all data underlying the findings in their manuscript fully available?

Reviewer #1: Yes

Reviewer #2: (No Response)

5. Is the manuscript presented in an intelligible fashion and written in standard English?

Reviewer #1: Yes

Reviewer #2: (No Response)

6. Review Comments to the Author

Reviewer #1: Figure 1, numbers along the colorbar are too small to read. Please try less labels and large font size

Figure 4, a color scale without unit for both T1 and ADC does not help. Please make horizontal one for each modality.

Reviewer #2: (No Response)

7. PLOS authors have the option to publish the peer review history of their article (what does this mean?). If published, this will include your full peer review and any attached files.

Reviewer #1: No

Reviewer #2: No

---

## [Editor Report · Acceptance letter]

16 Aug 2021

PONE-D-21-07115R1 

Development of a Standardized MRI Protocol for Pancreas Assessment in Humans 

Dear Dr. Virostko:

I'm pleased to inform you that your manuscript has been deemed suitable for publication in PLOS ONE. Congratulations! Your manuscript is now with our production department. 

Kind regards, 

on behalf of

Dr. Mingwu Jin 

Academic Editor

PLOS ONE